# Blood Endothelial-Cell Extracellular Vesicles as Potential Biomarkers for the Selection of Plasma in COVID-19 Convalescent Plasma Therapy

**DOI:** 10.3390/cells11193122

**Published:** 2022-10-04

**Authors:** Nada Amri, Nolwenn Tessier, Rémi Bégin, Laurent Vachon, Philippe Bégin, Renée Bazin, Lionel Loubaki, Catherine Martel

**Affiliations:** 1Faculty of Medicine, Université de Montréal, Pavillon Roger-Gaudry, 2900 Edouard Montpetit Blvd, Montreal, QC H3T 1J4, Canada; 2Montreal Heart Institute, 5000, Belanger Street, Montreal, QC H1T 1C8, Canada; 3Department of Pediatrics, CHU Sainte-Justine, 3175 Chemin de la Côte-Sainte-Catherine, Montreal, QC H3T 1C5, Canada; 4Department of Medicine, Centre Hospitalier de l’Université de Montréal, 900, Rue Saint-Denis, Montreal, QC H2X 0A9, Canada; 5Medical Affairs and Innovation, Héma-Québec, 1070, Avenue des Sciences-de-la-Vie, Québec, QC G1V 5C3, Canada

**Keywords:** COVID-19, convalescent plasma therapy, blood endothelium, extracellular vesicles, biomarkers

## Abstract

Despite the advancement of vaccination and therapies currently available, deaths due to the coronavirus disease 2019 (COVID-19) are still heavily documented. Severely infected individuals experience a generalized inflammatory storm, caused by massive secretion of pro-inflammatory cytokines that can lead to endothelial dysfunction, cardiovascular disease, multi-organ failure, and even death. COVID-19 convalescent plasma (CCP) therapy, selected primarily based on anti-SARS-CoV-2 antibody levels, has not been as convincing as expected in the fight against COVID-19. Given the consequences of a dysfunctional endothelium on the progression of the disease, we propose that the selection of plasma for CCP therapy should be based on more specific parameters that take into consideration the effect on vascular inflammation. Thus, in the present study, we have characterized a subset of CCP that have been used for CCP therapy and measured their anti- or pro-inflammatory effect on human coronary artery endothelial cells (HCAECs). Our data revealed that the longer the time lapse between the onset of symptoms and the plasma donation, the more mitochondrial dysfunction can be evidenced. The concentration of blood endothelial cell extracellular vesicles (BEC-EVs) was increased in the plasma of young individuals with mild symptoms. This type of selected convalescent plasma promoted the activation of the blood vascular endothelium, as reflected by the overexpression of *ICAM1* and *NFκB1* and the downregulation of VE-Cadherin. We propose this mechanism is a warning signal sent by the injured endothelium to trigger self-defense of peripheral blood vessels against excessive inflammation. Therefore, these results are in line with our previous data. They suggest that a more specific selection of COVID-19 convalescent plasma should be based on the time of donation following the onset of the clinical symptoms of the donor, the severity of the symptoms, and the age of the donor. These characteristics are relatively easy to identify in any hospital and would reflect the concentration of plasma BEC-EVs and be optimal in CCP therapy.

## 1. Introduction

The number of individuals affected by the severe acute respiratory syndrome coronavirus 2 (SARS-CoV-2) is increasing exponentially, and the disease remains a challenge to humanity [1]. Despite the advancement of general vaccination and the development of new therapeutic and preventive options, some individuals are still infected and hospitalized following the inflammatory storm [2]. Blood endothelial damage following the cytokine storm is also experienced by patients severely infected with COVID-19 [3,4,5]. Endothelial dysfunction leads to the development of chronic inflammatory diseases [6,7], as well as serious complications including acute respiratory distress syndrome (ARDS) and acute coronary syndrome. Furthermore, the vast majority of mortalities following COVID-19 infection are due to the cytokine storm and multi-organ failure subsequent to endothelial dysfunction [8]. Thus, it is essential to understand the mechanisms underlying endothelial overactivation following SARS-CoV-2 infection.

COVID-19 convalescent plasma (CCP) therapy has been thought to prevent these consequences and potentially reduce the risk of mortality. However, proper selection of CCP remains essential to optimize its beneficial effect while limiting its detrimental consequences on the hospitalized patient. Given the impact of a dysfunctional endothelium on the progression of inflammatory diseases [9,10], proposed treatments to counteract the harmful complications of COVID-19 infection should ideally consider the effect on the vascular network. The safety and efficacy of COVID-19 convalescent plasma therapy has also been tested in several clinical trials [11,12]. Yet its effects on the integrity of the vascular endothelium are not well understood. In a previous publication, we focused on identifying COVID-19 convalescent plasma with the least deleterious impact on lymphatic endothelial cells [13]. The lymphatic network is essential for proper clearance of pro-inflammatory mediators from host tissues and organs [14]. We proceeded to identify a particular signature of CCP that could be beneficial for the integrity of the lymphatic endothelium. Our results showed that plasma issued from early donations of CCP could protect against the loss of lymphatic endothelial cell integrity [13]. Conversely, late donations of CCP could promote loss of lymphatic endothelial integrity, and accentuate the permeability of the lymphatic endothelium. Dysfunctional lymphatic drainage could impair the clearance of immune cells and pro-inflammatory mediators, leading to the exacerbation of inflammation in patients infected with COVID-19. Extracellular vesicles (EVs) are released by cells during their activation or death [15]. Given their importance as biomarkers in several pro-inflammatory diseases, we characterized their presence in CCP. EVs derived from lymphatic endothelial cells (LEC-EVs) were abundantly found in early donated convalescent plasma. We thus proposed that their secretion by an impaired endothelium may be a warning signal that triggers the self-defense of peripheral lymphatic vessels against excessive inflammation.

Taking into consideration the impact of a dysfunctional endothelium on the progression of COVID-19, we now seek out to investigate whether this model remains true for blood endothelial cells. Thus, in the present study, we sought to identify the most adapted plasma to be used for CCP therapy that would limit vascular inflammation in the hospitalized recipient patients. Emphasizing on one of the endpoints of vascular activation and inflammation, we herein assessed the plasma content of blood endothelial cell extracellular vesicles (BEC-EVs) from a heterogenous group of plasma donors. Subsequently, we measured the impact of CCP on the integrity of human coronary artery endothelial cells (HCAECs). Altogether, we aimed to identify new and easily identifiable parameters to optimize the selection of CCP and limit consequences to the hospitalized recipient patient.

## 2. Materials and Methods

### 2.1. Study Participants

This study included 45 plasma donors who had recovered from COVID-19 and who participated in the pan-Canadian clinical study CONCOR-1 (Convalescent Plasma for COVID-19 Respiratory Illness) [16]. All donors included in this study gave written informed consent to participate in the CONCOR-1 trial and were recruited between 24 April and 12 July 2020. The present project has been approved by the Montreal Heart Institute Ethic Committee (protocol #2021-2812) and Héma-Québec (protocol #2020-004), in accordance with the Declaration of Helsinki. Women with previous pregnancies were excluded from donation. All participants had documented SARS-CoV-2 infection as assessed by a positive reverse transcription-polymerase chain reaction test, and human anti-SARS-CoV-2 Receptor-Binding-Domain (RBD) antibodies were quantified as previously published [17]. A minimum of 14 days after clinical recovery from COVID-19 were required before convalescent plasma collection. The time of donation was defined by the time lapse between the onset of symptoms and the convalescent plasma donation. The severity of the symptoms was quantified using a questionnaire answered by donors and graded as follows: 0 - asymptomatic, 1 - mild, 2 - moderate, and 3 - severe. None of the 45 participants included in our study had been hospitalized due to COVID-19. The donors were aged between 18 and 69 years old, and 38% were women. Plasma donation (500–700 mL) was made at 67 days on average after the onset of symptoms [13]. Plasma was divided into aliquots and frozen at −80 °C until use. The collection and distribution of the samples was performed and administered by Héma-Québec.

### 2.2. Cell Culture

Human coronary artery endothelial cells (Lonza cat. CC-2585) were seeded in complete endothelial cell growth medium MV2 (EGM-MV-2, PromoCell, cat. C-39221), which is a low-serum (5% *V/V*) medium. The cells were used at a passage of 4 to 6. Cells were equilibrated and treated with 10% convalescent plasma for 4 h in heparin-supplemented endothelial basal medium (EBM). EBM or fibrinogen- and SARS-CoV-2-free plasma (SeraConTM II Negative Diluent) was used as control. Interleukin 6 (IL-6, 20 ng/mL, PeproTech, cat. 10778-280), tumor necrosis factor alpha (TNF⍺, 20 ng/mL, R&D Systems, cat. 210-TA-10), and interferon gamma (IFNɣ, 10 ng/mL, PeproTech, cat. 10773-476) were then added on the endothelial monolayer for 20 h to induce inflammation. HCAECs maintained in EGM-MV-2 for the whole 24 h were used as control.

### 2.3. Measurement of Cellular Viability

Cell death was assessed as previously described [13,18]. Briefly, HCAECs were stained with annexin V (BD Biosciences, cat. 560506) and propidium iodide (PI) (Biotium, cat. 40017) and analyzed by flow cytometry (BD FACS CelestaTM). FlowJoTM version 10 was used for data processing. Gating strategies are depicted in Appendix A.

### 2.4. Analyses of Extracellular Vesicles in Plasma

Plasma EVs in CCP were identified and quantified as previously detailed [13,19,20]. Briefly, a small particles option flow cytometer (BD FACS CelestaTM) was used to identify EVs. The 450/40 bandpass filter (BV421, violet laser) was manually swapped with a 1 mm-thick magnetron sputtered 405/10 bandpass filter (Chroma Technology, Bellows Falls, VT, USA), which is referred to as V-SSC in Appendix A. The flow cytometer was first calibrated for EV detection using the ApogeeMix (#1493, Apogee Flow Systems, Hemel Hempstead, UK), a mixture of non-fluorescent silica beads (180, 240, 300, 590, 880, and 1300 nm) and FITC-fluorescent latex beads (110 and 500 nm), and count beads (Apogee Flow System cat. #1426). Samples were stained with carboxyfluorescein succinimidyl ester (CFSE) (Cedarlane, cat. S8269-10MG, 1/5000) to identify vesicles derived from cells. CFSE needed to be converted by active esterases to fluoresce, suggesting that gated events in the first panel of Appendix A were derive from cells. The anti-major histocompatibility complex 1 (MHC-I) (BD Biosciences, cat. 555553, dilution 1/100), anti-CD45 (BD Biosciences, cat. 560777, 1/200), and anti-CD62e (BD Biosciences, cat. 563359, 1/75) antibodies were used to select the specific subsets of EVs. FACS plots and histograms show all parameters in height (indicated as –H), as recommended for EV detection [21]. The threshold for the FSC detector was set at 200 V in FACS Diva software (BD Biosciences).

### 2.5. Messenger RNA Analysis by RT-qPCR

HCAECs were harvested, suspended in RNA Extraction Reagent, and stored at −80 °C before RNA extraction and qPCR analysis as previously described [13]. The primers used are the following: *ACTB* (F: ACGACATGGAGAAAATCTG and R: ATGATCTGGGTCATCTTCTC) *ICAM1* (F: ACCATCTACAGCTTTCCG and R: TCACACTTCACTGTCACC), *NFκB1* (F: CACAAGGAGACATGAAACAG and R: CCCAGAGACCTCATAGTTG), *NDFUA9* (F: CATTTCCGGAAGCCATTATC and R: CATCTACGACATATACTGGTTG), and *UQCRC2* (F: GTGAGTCATCCTGTTCTAAAG and R: CATTCTGTTCTCGGATTTCAC). Amplification plots are presented in Appendix A.

### 2.6. Immunoblotting

Immunoblotting was performed on HCAECs as previously described [13]. Anti-VE-Cadherin (Abcam, cat. Ab33168) and anti-phospho-VE-Cadherin (Invitrogen, cat. 44-1144G) antibodies were used. Original blots are presented in Appendix A.

### 2.7. Statistical Analysis

Associations between variables were calculated by Spearman’s rho or Pearson’s correlation, as appropriate. Non-parametric parameters were log transformed as indicated. Normality was assessed by the Shapiro-Wilk test. Analyses were performed using SPSS version 27 software and figures made with Prism version 9 software (GraphPad). *p* value ≤ 0.05 was defined as statistically significant.

## 3. Results

### 3.1. An Early Donation of Convalescent Plasma Increases the Expression of Mitochondrial Genes by Blood Endothelial Cells

We first investigated whether the anti-SARS-CoV-2 antibody concentrations and the duration of symptoms of the donors were associated with deterioration of the blood vascular endothelium. For this purpose, CCP was incubated on a monolayer of HCAECs for 4 h. Cell death was assessed by flow cytometry, following annexin V and propidium iodide labeling. Our results show that high plasma concentration of anti-SARS-CoV-2 antibodies did not correlate with an increase in cells undergoing late apoptosis, characterized by annexin V^+^ and PI^+^ labeling (r = 0.214, *p* = 0.463) (Figure 1A). Furthermore, the duration of symptoms of donors did not correlate with cell death (r = −0.180, *p* = 0.538) (Figure 1B). Lastly, the severity of the symptoms did not correlate with late apoptosis nor necrosis of blood endothelial cells (r = 0.102, *p* = 0.805) (Figure 1C).

Next, we tested whether the time of donation since the onset of the symptoms influenced blood endothelium integrity and propensity to inflammation. We found that blood endothelial cells expressed more NADH dehydrogenase 1 alpha subcomplex subunit 9 (NDUFA9) and cytochrome b-c1 complex subunit 2 (UQCRC2) by quantitative PCR when incubated with early donated convalescent plasma (Figure 1D,E). Thus, these two genes were less expressed when incubated with late donated convalescent plasma. A decrease in the expression of these genes could lead to mitochondrial dysfunction, resulting in a loss of function of the blood endothelial cells. This reflects the beneficial impact of incubating a CCP donated early following the resolution of symptoms on the blood endothelium.

### 3.2. Convalescent Plasma with High Concentrations of Extracellular Vesicles Derived from Blood Endothelial Cells Increases the Activation of Human Coronary Artery Endothelial Cells

In the subsequent experiment, we measured parameters that are reflecting endothelial cell activation following the incubation of convalescent plasma on cultured HCAECs. Our results show that BEC exposed to plasma that contains low BEC-EVs underwent more late apoptosis or necrosis, as measured by annexin V and PI labeling (r = −0.564, *p* = 0.045) (Figure 2A). In addition, plasma rich in BEC-EVs induced an increased expression of *ICAM1* mRNA and nuclear factor-kappa B subunit 1 (*NFκB1*) by HCAECs (Figure 2B,C). Moreover, higher concentrations of BEC-EVs led to a decrease in the expression of the phosphorylated form of VE-Cadherin (Figure 2D). Whereas BEC-EVs could represent a reliable biomarker in the selection of the optimal CCP for CCP therapy, we are aware that not all clinical laboratories worldwide can assess the presence of extracellular vesicles. Therefore, we sought to identify universal parameters that could reflect the presence of this specific subset of EVs.

### 3.3. Extracellular Vesicles Derived from Blood Endothelial Cells Are More Abundant in Convalescent Plasma from Donors with Mild Symptoms

We then sought to identify which convalescent plasma was associated with a high concentration of BEC-EVs. We found that plasma from patients with mild symptoms (n = 15) contained more BEC-EVs compared with patient with moderate (n = 9) or severe symptoms (r = −0.397, *p* = 0.024; n = 7) (Figure 3A). We also observed that BEC-EVs were more abundant in younger patients (r = −0.451, *p* = 0.001) (Figure 3B). Finally, our data revealed that patients with low plasma anti-SARS-CoV-2 antibody concentration had higher concentrations of BEC-EVs (r = −0.332, *p* = 0.021) (Figure 3C). However, using a multivariate model, collinearity between the severity of symptoms and age of the donors was observed. In the multivariate model, severity of symptoms was the only variable independently associated with BEC-EVs levels (β = −0.456, *p* = 0.018). Including the concentration of anti-SARS-CoV-2 antibodies, we failed to reach statistical significance for this variable (β = −0.163, *p* = 0.377).

## 4. Discussion

Convalescent plasma therapy has been extensively studied since the beginning of the COVID-19 pandemic [11]. In most of these studies, the criterion for the selection of CCP was based on the plasma concentration of antibodies. Yet, CCP therapy has not been as efficient as expected. Therefore, we have sought to investigate whether other parameters can be considered when selecting the plasma used for CCP. In a recent study, we have demonstrated that early donations of CCP could be beneficial for the integrity of lymphatic endothelial cells [13]. Given the importance of the lymphatic system in the clearance of immune cells and other pro-inflammatory modulators from peripheral tissues, damage to its integrity could compromise this critical function [22]. Nevertheless, an adequate selection of CCP must also be beneficial for the integrity of the blood vascular system, given its significance in the development of diseases. The administration of inadequate CCP for the blood endothelium could be detrimental for the patient. Indeed, following incubation of plasma from severely infected patients with the virus on a pulmonary endothelium, Rauch et al. revealed that the subsequent endothelial dysfunction could be attributable to the increased presence of pro-inflammatory cytokines in the plasma [3]. This alteration in endothelial function following virus internalization is manifested by an imbalance between the bioavailability of vasodilators and vasoconstrictors. A decrease in the production of nitric oxide (NO), a potent vasodilator, and an increase in the production of ROS have been demonstrated [4]. This induces a state of oxidative stress for the endothelial cell, contributing to the progression of the pathogenesis of COVID-19 disease [4].

Our previous results showed that early donation of CCP could protect against lymphatic dysfunction [13]. Conversely, late donations of CCP could promote loss of endothelial integrity, as well as accentuate the permeability of the lymphatic endothelium. Herein, we first demonstrated that CCP incubated on cultured HCAECs did not impact cell death regardless of plasma anti-SARS-CoV-2 antibody concentration, duration, and severity of symptoms. Interestingly, we found, however, that blood endothelial cells expressed less *NDUFA9* and *UQCRC2* when incubated with late donations of convalescent plasma. NDUFA9 is an essential subunit for the assembly and stability of complex I, the first and largest complex of the mitochondrial respiratory chain [23]. This complex is divided into three functional fractions essential for oxidative phosphorylation (OXPHOS) [23]. A deficiency of this complex is one of the most common defects of the OXPHOS system [23]. Dysfunctions of this system can lead to several devastating diseases [24]. Recently, it has been shown that *NDFUA9* gene expression is decreased in mitochondria in placenta from asymptomatic and symptomatic women infected with COVID-19. This decrease is associated with the reduction in other mitochondrial genes, such as *SDHA*, *COX41,* and *UQCRC1,* suggesting an impairment of mitochondrial function during COVID-19 infection [25]. The UQCRC2 protein is essential for the function of complex III of the mitochondrial respiratory chain [26]. Enzyme deficiency of mitochondrial complex III is responsible for rare diseases such as Bjornstad syndrome, GRACILE syndrome, and Leigh syndrome, which can cause severe complications including hypoglycemia, lactic acidosis, ketosis, intrauterine developmental delay, and liver failure [26,27]. These data support that there is a mitochondrial dysfunction within blood endothelial cells and underline that mitochondria could be a potential target to protect the endothelium [28]. Therefore, early donated CCP is deemed beneficial for the blood endothelium.

Given the abundance of LEC-EVs previously found in early donated convalescent plasma [13], we proposed that their secretion by an impaired endothelium may be a warning signal that triggers the self-defense of peripheral lymphatic vessels against excessive inflammation. We hypothesized that BEC-EVs would have a similar effect. We found that the incubation of HCAECs with plasma low in BEC-EVs correlated with more cells undergoing late apoptosis or necrosis, proven to be harmful to cells [29]. In addition, high levels of plasma BEC-EVs correlated with higher production of HCAECs activation markers. This supports the hypothesis of EVs being produced as a warning signal, as we have previously shown [13]. Moreover, EVs from plasma of COVID-19 patients contain pro-inflammatory, pro-coagulation, and tissue-remodeling markers [30]. EVs could be used to deliver cytokines/chemokines to recipient cells and tissues [31]. Indeed, these BEC-EVs, produced in large quantities by the CCP donor, could be internalized by the host blood endothelial cells and in turn modify their function to express markers found during inflammation. This would prepare the recipient cells or tissues to adapt to the upcoming inflammation observed during COVID-19. In addition to the overexpression of *ICAM1* and *NFκB1*, we also observed a decrease in phosphorylated VE-Cadherin expression following the incubation with CCP containing a high concentration of BEC-EVs. During an inflammatory process, the blood endothelium increases the expression of certain adhesion molecules for the adhesion of monocytes and immune cells that can fight the pathogen [32]. However, these immune cells must sometimes extravasate through the endothelium [32], thus decreasing the adhesion of endothelial cells to each other, which explains this decrease in VE-Cadherin expression. A study by Gotsch and colleagues demonstrated that VE-Cadherin is involved in the process of leukocyte extravasation through the endothelium [33]. Following the injection of a monoclonal antibody blocking VE-Cadherin aggregation, they observed an increase in neutrophil migration across the endothelium [33]. In addition, neutrophil docking to the apical surface of cultured endothelial cells was shown to cause degradation of VE-Cadherin-associated β-catenin and led to an increase in the permeability of the endothelium for macromolecules [34]. Thus, one possible mechanism could be that CD45^-^CD62e^+^ EVs could adhere to HCAECs and be internalized to then increase the endothelium permeability and facilitating leukocyte extravasation.

## 5. Conclusions

Altogether, our data suggest that identifying convalescent plasma rich in BEC-EVs could help maintain the integrity of the blood endothelium and in turn promote the chances of patient remission. The concentration of BEC-EVs was herein increased in plasma isolated from young patient with mild symptoms. Based on previous results indicating that plasma issued from early post-infection donation should be prioritized to better protect the lymphatic endothelium, these additional data suggest that the age of the donor and the severity of the symptoms should also be taken in consideration when selecting CCP. These results demonstrate the importance of properly selecting convalescent plasma for therapeutic use to optimize its beneficial effects on the infected and hospitalized patient. This remains essential for future pandemics and/or epidemics where convalescent plasma therapy could potentially be beneficial.

## Figures and Tables

**Figure 1 cells-11-03122-f001:**
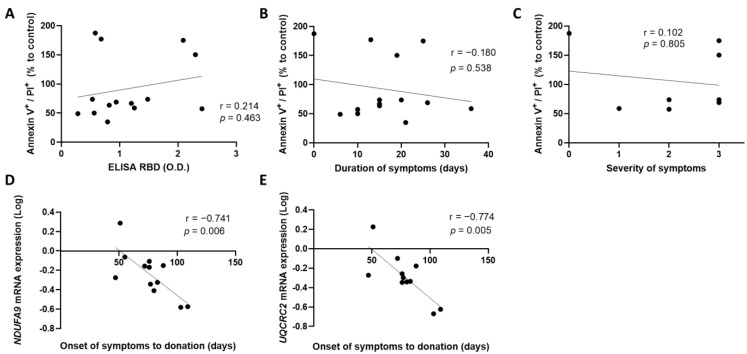
Clinical characteristics, cell death, and mitochondrial gene expression. Correlation between late apoptosis and (**A**) plasma anti-SARS-CoV-2 antibody concentration, (**B**) duration of symptoms of donors, and (**C**) severity of symptoms (from 0 to 3, 3 being the most severe) was assessed. The time from the onset of symptoms to donation was then correlated with the expression of (**D**) *NDUFA9* and (**E**) *UQCRC2* mRNA expressed by HCAECs. Data are represented as a relative percentage to cells treated with control plasma. Significance was determined by Spearman correlation. *p* ≤ 0.05 was considered significant. PI: propidium iodide; RBD: receptor binding domain; O.D.: optical density NDUFA9: NADH dehydrogenase 1 alpha subcomplex subunit 9; UQCRC2: cytochrome b-c1 complex subunit 2.

**Figure 2 cells-11-03122-f002:**
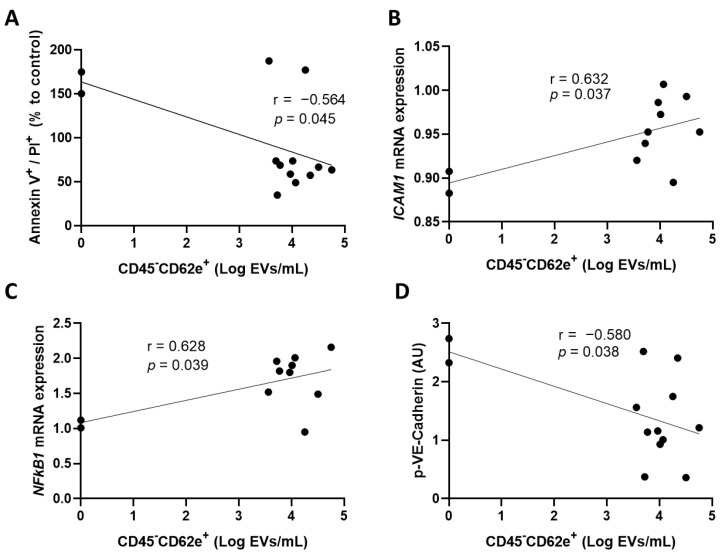
Circulating CD45^-^CD62e^+^ extracellular vesicles and endothelial cell activation. The correlation between CD45^-^CD62e^+^ EVs and (**A**) endothelial cells undergoing late apoptosis and necrosis, (**B**) endothelial cell *ICAM1* mRNA expression, (**C**) endothelial cell *NFκB1* mRNA expression; (**C**,**D**) phosphorylated VE-Cadherin protein expression was assessed. Data are represented as a relative percentage to cells treated with control, uninfected plasma. Significance was determined by Pearson correlation. *p* ≤ 0.05 was considered significant. PI: propidium iodide; EVs: extracellular vesicles; AU: arbitrary units.

**Figure 3 cells-11-03122-f003:**
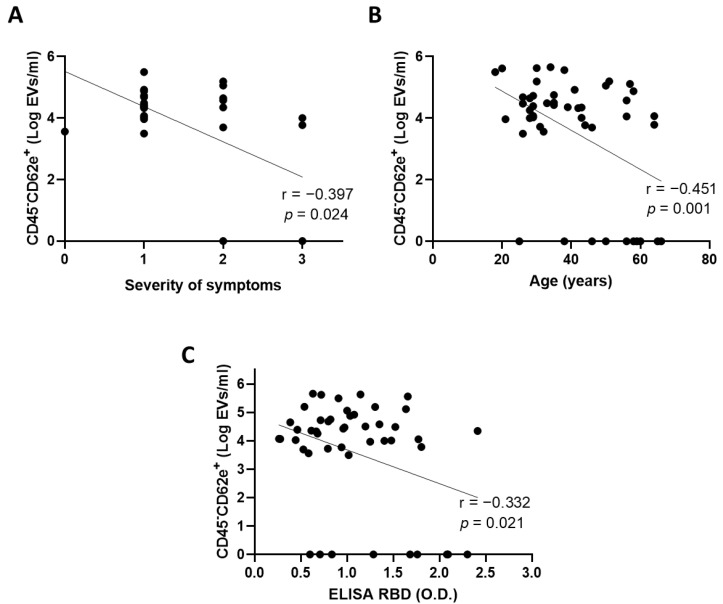
Circulating CD45^-^CD62e^+^ EVs and clinical parameters. The correlation between CD45^-^CD62e^+^ EVs and (**A**) the severity of symptoms of donors, (**B**) the age of donors, and (**C**) the concentration of plasma anti-SARS-CoV-2 antibodies was assessed. Significance was determined by a Spearman correlation for symptom severity and by a Pearson correlation for age and antibody concentration. *p* ≤ 0.05 was considered significant. Multiple linear regression was performed using the forward method, since more than one variable in the univariate analysis was associated with the outcome tested. EVs: extracellular vesicles; RBD: receptor binding domain; O.D.: optical density.

## Data Availability

Data are contained within the article.

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
