# Peer review of "Blood Endothelial-Cell Extracellular Vesicles as Potential Biomarkers for the Selection of Plasma in COVID-19 Convalescent Plasma Therapy"

_cells, 2022, doi:10.3390/cells11193122_

Round 1

Reviewer 1 Report

Congratulations on taking up a topic that I believe is important. The more we study, the more we know that the regulation of the immune response goes beyond the cytokine framework. Current data on the microRNA contained, inter alia, in extracellular vesicles, indicate another mechanism that regulates the functions of cells.

The title of the manuscript proposed for review already intrigued me scientifically. A well-written introduction, interesting concept and clinically important.

However, I have some comments on the rest of the manuscript:

1. The authors repeatedly cite the previous publication, in the introduction, methodology, and results, and again in the discussion. Perhaps this was the assumption to relate everything to previous results however, in the results section it is unacceptable. It is difficult to read which results are up-to-date. Only new results obtained should be included in this part of the work, without discussion! The assumptions are described in the introduction, and descriptions from this section are reproduced again in the discussion.

2. The methodological part does not provide details of the procedures, indicating the source of previous papers, not every reader has access to all publications, so I encourage you to indicate the methodology in this work.

3. The authors do not characterize the respondents, they indicate the source of the previous publication. In my opinion, the characteristics of the respondents in this work should be included, also due to the different number of results in the various presented studies. It is impossible to evaluate this work without the previous one, which makes it difficult for the reader to analyze the results. The question arises which studies were carried out in the entire group of 45 people and which in smaller ones. Were the groups analyzed by gender? Were the results of women analyzed due to pregnancy? What is early and late donation?

Author Response

Reviewer 1

Congratulations on taking up a topic that I believe is important. The more we study, the more we know that the regulation of the immune response goes beyond the cytokine framework. Current data on the microRNA contained, inter alia, in extracellular vesicles, indicate another mechanism that regulates the functions of cells.

The title of the manuscript proposed for review already intrigued me scientifically. A well-written introduction, interesting concept and clinically important.

However, I have some comments on the rest of the manuscript:

  1. The authors repeatedly cite the previous publication, in the introduction, methodology, and results, and again in the discussion. Perhaps this was the assumption to relate everything to previous results however, in the results section it is unacceptable. It is difficult to read which results are up-to-date. Only new results obtained should be included in this part of the work, without discussion! The assumptions are described in the introduction, and descriptions from this section are reproduced again in the discussion.

We thank the reviewer for these comments. We have modified the manuscript accordingly. 

  1. The methodological part does not provide details of the procedures, indicating the source of previous papers, not every reader has access to all publications, so I encourage you to indicate the methodology in this work.

We thank the reviewer for this remark. Albeit the editor mentioned that the method section was redundant with that of our previous article (despite paraphrasing), we now made sure to include insightful details in our methods. Of note, the previous publication is open access, so referred information is easily accessible.

  1. The authors do not characterize the respondents, they indicate the source of the previous publication. In my opinion, the characteristics of the respondents in this work should be included, also due to the different number of results in the various presented studies. It is impossible to evaluate this work without the previous one, which makes it difficult for the reader to analyze the results. The question arises which studies were carried out in the entire group of 45 people and which in smaller ones. Were the groups analyzed by gender? Were the results of women analyzed due to pregnancy? What is early and late donation?

We thank the reviewer for these comments. We have clarified these points. Please note that we did used the entire group of 45 people for this study. Pregnancy was a criterion of exclusion, so in this study no women were pregnant. Finally, we have referred to an ‘’early or late donation’’ when we considered the time between the onset of symptoms and the moment of the donation. We analyzed continuous variables, so no cut off has been set.

Reviewer 2 Report

The paper of Nada Amri et. al. presents an important research study on the quality control of convalescent plasma rich in BEC-EVs from COVID-19 infected patients.  The authors are bringing some important evidence that such controls would help maintain the integrity of the blood endothelium and in turn promote the chances of patient remission. The concentration of BEC-EVs was shown to be increased in plasma isolated from young patient with mild symptoms.  Aside of being required to be isolated from patients with early post-COVID-19 infection the CCP is proposed to be checked for the age of the donor and the severity of the symptoms associated with each donor.   

Before being accepted for publication the authors are required to submit the flow cytometry views for each staining performed to identity:

1) the BEC-EVs (for Circulating CD45- CD62e+ EVs) for the same patients for which the qPCR assay was run; 2) the annexin V and propidium iodide for showing late apoptosis. 

Please include the strategy of gating in each case.

In addition, please include the views of the western blots and graphs for qPCR for top 5 patients in the supplementary data.

Author Response

Reviewer 2

The paper of Nada Amri et. al. presents an important research study on the quality control of convalescent plasma rich in BEC-EVs from COVID-19 infected patients.  The authors are bringing some important evidence that such controls would help maintain the integrity of the blood endothelium and in turn promote the chances of patient remission. The concentration of BEC-EVs was shown to be increased in plasma isolated from young patient with mild symptoms.  Aside of being required to be isolated from patients with early post-COVID-19 infection the CCP is proposed to be checked for the age of the donor and the severity of the symptoms associated with each donor.  

Before being accepted for publication the authors are required to submit the flow cytometry views for each staining performed to identity:

1) the BEC-EVs (for Circulating CD45- CD62e+ EVs) for the same patients for which the qPCR assay was run; 2) the annexin V and propidium iodide for showing late apoptosis.

Please include the strategy of gating in each case.

We thank the reviewer for these comments. We have added supplemental figures that contain the flow cytometry views for the staining mentioned above (figure S1 and figure S2).

In addition, please include the views of the western blots and graphs for qPCR for top 5 patients in the supplementary data.

We thank the reviewer for this requirement. Figures S3 (for western blot) and S4 (for qPCR) have been added to the manuscript.